# A Systematic Review and Meta-Analysis on the Diagnostic and Prognostic Values of ^18^F-FDG PET in Uveal Melanoma and Its Hepatic Metastasis

**DOI:** 10.3390/cancers16091712

**Published:** 2024-04-28

**Authors:** Seyed Ali Mirshahvalad, Nazanin Zamani-Siahkali, Christian Pirich, Mohsen Beheshti

**Affiliations:** 1Division of Molecular Imaging and Theranostics, Department of Nuclear Medicine, University Hospital, Paracelsus Medical University, 5020 Salzburg, Austria; ali.mirshahvalad@uhn.ca (S.A.M.); nazaninzamanisiahkali@gmail.com (N.Z.-S.); c.pirich@salk.at (C.P.); 2Joint Department of Medical Imaging, University Health Network, Mount Sinai Hospital & Women’s College Hospital, University Medical Imaging Toronto (UMIT), University of Toronto, Toronto, ON M5G 2N2, Canada; 3Research Centre for Nuclear Medicine, Department of Nuclear Medicine, Shariati Hospital, Tehran University of Medical Sciences, Tehran 1461884513, Iran

**Keywords:** fluorodeoxyglucose, FDG, positron emission tomography, uveal, melanoma, metastasis, sensitivity, specificity, review, meta-analysis

## Abstract

**Simple Summary:**

Melanoma is the most common primary intra-ocular cancer in adults. In case of suspected extra-ocular tumoral involvement, patients may be required to undergo computed tomography (CT) and magnetic resonance imaging (MRI). Positron emission tomography (PET) with ^18^F-Fluorodeoxyglucose (^18^F-FDG) is another imaging modality that can be used in uveal melanoma assessment. Although comprehensive systematic reviews and meta-analyses were published on the role of ^18^F-FDG PET imaging in cutaneous melanoma, there is no ample evidence regarding its value in uveal melanoma patients. Thus, in this systematic review and meta-analysis, we tried to investigate the existing literature and provide a comprehensive study on the diagnostic and prognostic values of ^18^F-FDG PET in uveal melanoma.

**Abstract:**

In this systematic review and meta-analysis (PRISMA-compliant), we tried to investigate diagnostic and prognostic values of ^18^F-FDG PET in uveal melanoma. A systematic search was conducted on the main medical literature databases to include studies that evaluated ^18^F-FDG PET as the imaging modality to evaluate patients with uveal melanoma. Overall, 27 studies were included. Twelve had data about the detection rate of ^18^F-FDG PET in primary intra-ocular tumours. The pooled sensitivity was 45% (95%CI: 41–50%). Furthermore, studies showed that the larger the primary tumour, the higher its uptake. Among the included studies, 13 assessed ^18^F-FDG PET in detecting metastasis. The pooled sensitivity and specificity were 96% (95%CI: 81–99%) and 100% (95%CI: 94–100%), respectively. Regarding liver metastasis, they were 95% (95%CI: 79–99%) and 100% (95%CI: 91–100%), respectively. Noteworthy, the level of ^18^F-FDG uptake was a strong predictor of patient survival. Lastly, ^18^F-FDG PET could characterise lesions from the histopathology perspective, distinguishing high-risk from low-risk diseases. Overall, although not reliable in detecting primary intra-ocular tumours, ^18^F-FDG PET is highly accurate for diagnosing metastatic uveal melanomas. It can also be a highly valuable modality in terms of patient prognostication. Thus, ^18^F-FDG PET can be recommended in patients diagnosed with uveal melanoma to enhance decision-making and patient management.

## 1. Introduction

Although malignant tumours of the eye are not common, they can significantly affect patients’ quality of life and survival, as well as being challenging in terms of patient management [1]. Melanoma is the most common primary intra-ocular cancer in adults [2]. It comprises 5% of all melanomas. Uveal melanoma is mainly detected during ophthalmic examination and can originate from various structures, including the iris, ciliary body, and choroid [3].

Its initial assessment typically includes fundus photography, fluorescein angiography, fundus autofluorescence, optical coherence tomography, or ultrasound imaging. However, in cases of suspected extra-ocular tumoral involvement, patients may be required to undergo computed tomography (CT) and magnetic resonance imaging (MRI) [1,4]. Although at the time of diagnosis, metastases are detected in less than 5% of patients with uveal melanomas, more than half eventually develop metastases, the liver being the first site of metastasis in the majority (about 90%) of patients [4,5,6].

Positron emission tomography (PET) with ^18^F-Fluorodeoxyglucose (^18^F-FDG) is another imaging modality that is widely used in uveal melanoma patients. It is not currently a routine imaging in patient management, though it may help in diagnosis, staging, and prognostication [6,7]. It was reported to be more likely to detect both extrahepatic and hepatic metastases when used for initial staging compared to other whole-body radiographic imaging [8]. ^18^F-FDG PET shows the metabolic activity of the tumours based on their glucose consumption at the cellular level, and its utilization is well-established in other types of melanomas [6,9].

Although comprehensive systematic reviews and meta-analyses were published on the role of ^18^F-FDG PET imaging in non-ophthalmic melanoma [9,10], there is no ample evidence regarding its value in uveal melanoma patients. Thus, in this systematic review and meta-analysis, we tried to investigate the existing literature thoroughly and provide a comprehensive study on the diagnostic and prognostic values of ^18^F-FDG PET in patients with uveal melanoma.

## 2. Methodology

This study was designed following the Preferred Reporting Items for Systematic Review and Meta-Analysis of Diagnostic Test Accuracy (PRISMA-DTA) studies protocol [11].

### 2.1. Search Strategy

A systematic search was conducted on the three main medical literature databases, Scopus, Web of Science, and PubMed, up to 4 January 2024. The search was conducted using database-specific search strategies, covering all key terms (including MeSH terms) to include the mainstem search strategy: ((“ophthalm*” OR “uvea*” OR “choroid*” OR “orbit*” OR “ocular” OR “eye”) AND (“pet*” OR “positron” OR “*fdg*”) AND “melanoma”)]. No time or language limit was considered for the search. The electronic searches were augmented by searching through the references in the included articles to add any other possible relevant study.

### 2.2. Study Selection

Published original articles that met the following inclusion criteria were considered eligible for inclusion in this study:(1)Studies that evaluated ^18^F-FDG PET or PET/CT as the imaging modality to evaluate patients with uveal melanoma. For the sake of this review, we will refer to ^18^F-FDG PET and PET/CT with the general terminology of ^18^F-FDG PET in the rest of the manuscript;(2)In the case of diagnostic performance, studies that provided information regarding the detection of tumoral lesions by ^18^F-FDG PET, considering both primary intra-ocular lesion and metastasis. Additionally, studies that provided data regarding the correlation of ^18^F-FDG PET-derived parameters with tumour size were discussed in the diagnostic section;(3)In the case of prognostic evaluation, studies that followed the standard survival analyses and provided hazard ratios for their calculations. The studied prognosticators should be derived from ^18^F-FDG PET scans to be included.

Notably, for eligibility to be included in the meta-analytic calculations, studies should provide adequate data to calculate true-positive, true-negative, false-positive, or false-negative results. These data were required for sensitivity or specificity calculations.

The exclusion criteria included meeting abstracts, abstracts without full articles, and unpublished/not peer-reviewed articles. Duplicated studies were excluded. The titles and abstracts were reviewed to check the studies’ eligibility to enter the full-text review phase. Two reviewers (NZS and SAM with four and five years of expertise in molecular imaging, respectively) independently screened titles and abstracts separately to identify relevant citations. In case of any discrepancy, a third reviewer (MB; molecular imaging expert) decided on the eligibility.

### 2.3. Data Extraction

For the remaining studies, data extraction was performed by two reviewers (NZS and SAM) after the full-text review. Extracted data included the title, first author’s name, the aim of the study, modality, patient population, age, histopathology details, status of the evaluated patients (initial assessment versus post-treatment evaluation), treatment-naïve patient percentage, male percentage, location of the primary lesions (uveal versus choroidal melanoma), primary tumour mean thickness/height, primary tumour mean diameter, primary tumour average SUVmax, and the average SUVmax of metastases. Next, diagnostic and prognostic information were gathered.

### 2.4. Statistical Analysis (The Meta-Analytic Parts)

The retrieved data were presented as either means or percentages, depending on whether the variables were continuous or categorical. To analyze the pooled diagnostic performance, the hierarchical method was employed to pool the random effect model’s measures of sensitivity, specificity, positive likelihood ratio, and negative likelihood ratio from the derived two-by-two contingency tables. The pooled measurements were calculated to detect primary intra-ocular tumours, metastatic diseases, and hepatic metastasis. The bivariate model was employed to determine sensitivity and specificity, along with their corresponding 95% confidence intervals (CI), to account for any variations within and across studies. Additionally, a scattergram was created for hepatic metastasis detection to analyze the overall results. The analyses were conducted using the STATA 16 (StataCorp) software modules “Midas” and “Metaprop” [12,13].

## 3. Results

Overall, 27 studies were included in this systematic review and meta-analysis [14,15,16,17,18,19,20,21,22,23,24,25,26,27,28,29,30,31,32,33,34,35,36,37,38,39,40]. Their detailed characteristics can be found in Table 1. The study selection flowchart is provided in Figure 1.

### 3.1. Intra-Ocular Evaluation

Among the included studies, 12 had data about the detection rate of ^18^F-FDG PET in diagnosing primary intra-ocular tumours in patients with suspected/proven melanoma. The pooled sensitivity of these studies was 45% (95%CI: 41–50%). The forest plot can be found in Figure 2.

Notably, Papastefanou et al. [37] reported two thresholds for primary tumour detection, SUVmax > 2.5 and >4. In our meta-analytic calculation, we used the SUVmax threshold > 4 since the former seemed more of an outlier (sensitivity of 93% in 76 patients) compared to the existing literature. Having said that, studies like McCannel et al. [34] and Reddy et al. [38] also used SUVmax > 2.5 but reported more *realistic* detection rates of 19% and 28%, respectively. Moreover, in the study by Calcagni et al., they considered all uveal lesions as ^18^F-FDG PET-positive when they showed ^18^F-FDG uptake independent of their uptake value (no SUV cut-off), making their detection rate as high as 88% [14]. Furthermore, if we applied the threshold of >4 instead of >3 in the study of Matsuo et al., their reported sensitivity would drop significantly from 71% to 43% [33].

Also noteworthy is that two studies reported the specificity for ^18^F-FDG PET for primary tumour detection, being 100% in both [24,25].

### 3.2. Association between ^18^F-FDG Uptake and Primary Tumour Characteristics

Multiple studies evaluated the correlation between the metabolic uptake of the primary tumour and its characteristics, particularly in terms of lesion size. The majority of the studies supported that the larger the primary tumour, the higher the uptake. Faia et al. and McCannel et al. reported that ^18^F-FDG uptake was positively correlated with tumour thickness/height and diameter [18,34]. Lee et al. also showed that tumour thickness was significantly greater in the metabolically active lesions than in the inactive ones [31]. They noted avid tumours had greater choroidal thickness, particularly in the sub-foveal region. Modorati et al. also supported the association between the tumour diameter and ^18^F-FDG PET positivity [35]. Papastefanou et al. investigated more parameters and showed that SUVmax was positively correlated with tumour thickness, area and volume [37]. Lastly, Reddy et al. reported that in their investigation, no small (T1) tumour showed avidity, but 33% of medium-sized (T2) melanomas and 75% of large (T3) tumours revealed ^18^F-FDG PET uptake, supporting the association of avidity with size [38].

Furthermore, Finger et al. showed that lesions with higher uptake were larger and taller [20]. Also, they found that the majority of highly-avid tumours were mushroom-shaped. In their later study [21], Finger et al. investigated how much tumour regression could affect SUV. On average, tumours regressed to 50% of their initial height after radiation therapy and consequently showed a significant decrease in their uptake value. Lee et al. also found that the median tumour thickness at presentation was higher in metabolically active tumours compared to inactive lesions. Similar to Finger et al., they assessed tumour regression over time but from another standpoint and reported that the percentage decrease in tumour size during early follow-up was significantly greater in metabolically active tumours than in inactive ones.

Having said that, in the study by Calcagni et al. [14], the size of the primary lesion was not correlated with ^18^F-FDG uptake. Also, Orcurto et al. reported that there was no correlation between SUV and the height or diameter of primary tumours in their study [36]. Additionally, Matsuo et al. did not find a significant correlation between SUVmax and the longest diameter or the circumference of the tumour, though SUVmax was positively correlated with tumour thickness [33]. Also, in the aforementioned study by Papastefanou et al., although various variables were significantly associated with SUVmax, the maximal diameter did not reach statistical significance [37].

### 3.3. Detecting Extra-Ocular Metastasis

Among the included studies, 13 investigated the diagnostic value of ^18^F-FDG PET in detecting melanoma metastasis. The pooled sensitivity and specificity were 96% (95%CI: 81–99%) and 100% (95%CI: 94–100%), respectively. Figure 3 shows the corresponding forest plots. Notably, as shown, only three datasets were related to regional metastasis, making our findings for this particular purpose insufficient to assess exclusively. Excluding databases regarding regional metastasis (focusing only on distant metastasis detection), the pooled sensitivity and specificity were 95% (95%CI: 80–99%) and 100% (95%CI: 89–100%), respectively.

### 3.4. Detecting Hepatic Metastasis

Regarding liver metastasis (the most common distant metastasis) in particular, there was well-established literature to be discussed further. The pooled sensitivity and specificity were 95% (95%CI: 79–99%) and 100% (95%CI: 91–100%), respectively. Figure 4a,b shows the corresponding forest plots and the likelihood ratio scattergram. As can be seen in the scattergram, ^18^F-FDG PET could accurately exclude or confirm hepatic metastasis in uveal melanoma patients.

There were also some prominent findings to mention in the included studies. Del Carpio et al. showed that all subcentimetric hepatic metastasis in their cohort were iso-metabolic compared with the normal hepatic uptake (ratio < 1.1) [16]. Orcurto et al. also reported that there was a strong correlation between the size of hepatic metastases on MRI and their SUV (SUVmax and lesion-to-liver uptake ratio) [36]. Subcentimetric metastases showed significantly lower SUV than other lesions, and only a few showed higher ^18^F-FDG uptake than the liver. However, they could also detect some ^18^F-FDG-positive tumoral lesions with significantly elevated SUVs in the liver that did not have MRI correlates, though these PET-only-positive lesions were only 4% of all detected metastases. Also noteworthy from their study is that although ^18^F-FDG PET could find at least one hepatic metastasis in each metastatic patient (accurate *patient-level* accuracy), it missed multiple lesions detected by MRI, mostly the small-sized metastases. This lower accuracy at the lesion level compared to the patient level was also supported by Servois et al. [39]. They showed that although ^18^F-FDG had a higher positive predictive value contrary to MRI (MRI showed one false-positive lesion), the lesion-level sensitivity of ^18^F-FDG PET was 45% compared to that of 67% in MRI, mainly due to missing subcentimetric hepatic metastases (MRI had poor sensitivity in <5 mm metastases).

Noteworthy, three of the included studies [19,23,33] were exclusive on choroidal melanoma in terms of the detection of hepatic metastasis. No false-negative results were reported in this patient population (sensitivity of 100%), with only one false-positive finding (specificity of 98–100%).

### 3.5. Other Diagnostic Values

Cohen et al. showed that ^18^F-FDG PET could reveal second synchronous malignancies in 10% of their study population, signifying the value of this modality as a one-stop-shop imaging [15]. Their found malignancies included lung, breast, colon, thyroid, and adrenal gland. Freton et al. also reported this value of ^18^F-FDG PET, helping to find a second primary malignancy (lung, colon, thyroid, breast, and lymphoma) in 10 (3%) of their patients [23].

### 3.6. Prognostic Value

Regarding the prognostic value of ^18^F-FDG PET findings in the primary tumour, Lee et al. reported that primary tumour SUVmax was a strong predictor of metastatic death (hazard ratio of 3.3), as well as the tumour’s largest diameter (hazard ratio of 1.7) [29]. In their analysis, post-treatment SUVmax > 2.2 was 71% sensitive and 88% specific in predicting metastatic death. Additionally, although with only borderline statistical significance, there was an inverse correlation between the initial SUV and time to metastasis. Considering the histopathology subtype as a tumour characteristic to determine prognosis, Faia et al. showed that the mixed cell had a significantly higher ^18^F-FDG avidity than other subtypes [18]. Calcagni et al. showed that SUVmax could distinguish mixed cell and epithelioid cell (high-risk subtypes) from spindle cell (low-risk subtype), SUVmax > 4.16 being in favour of high-risk disease [14]. Moreover, McCannel showed that there was a significant correlation between metabolic activity on ^18^F-FDG PET images and chromosome 3 loss, being a highly specific indicator [34]. This was similar to the finding of Papastefanou et al. [37]. They also mentioned that SUVmax was significantly higher in tumours with monosomy 3 compared to those with disomy 3. However, regarding the status of chromosome 8 in their study, there was no difference in SUVmax [*Although not within the realms of this study, it is good to note that monosomy 3 is associated with a higher risk of metastatic disease and mortality [41,42]. This fact may provide a better understanding of the value of ^18^F-FDG PET in patient prognosis.*].

In patients with hepatic metastasis, Del Carpio et al. mentioned that although ^18^F-FDG PET had false-negative results in six patients in terms of detecting hepatic metastasis, these patients with undetectable hepatic metastases showed favourable overall survival in the follow-up, significantly higher than those with true-positive hepatic metastases (overall survival of 58 months versus 17 months in the false-negative patients versus whole metastatic population, respectively) [16]. They also reported that SUVmax and SUVmax-to-liver ratio of hepatic metastases were significant predictors of overall survival. Interestingly, with an SUVmax of <8.5 (median SUVmax of their population), no death was documented in the first year, while the median overall survival in other patients (SUVmax ≥ 8.5 in hepatic metastases) was only 9 months. Furthermore, at different time points, the overall survival of patients with highly avid hepatic metastases was significantly lower than the others, 23% and 11% at 2 and 4 years versus 68% and 35%, respectively. Moreover, they showed that the median overall survival of patients with a high uptake ratio (SUVmax-to-liver ratio ≥ 1.86) was significantly lower than others, 13 months versus 38 months. Notably, SUVmax was a significant independent predictor of overall survival in their multivariate analysis (hazard ratio of 2.6 for SUVmax ≥ 8.5) along with the abnormal gamma-glutamyl-transferase level and the diameter of the largest metastasis (≥3 cm; M1b and M1c patients). In another study, Eldredge-Hindy et al. showed that metabolic tumour volume (MTV) and total lesion glycolysis (TLG) of hepatic metastases were significantly correlated with both hepatic progression-free survival and overall survival after microsphere brachytherapy [17]. Patients with low pre-treatment TLG (<225) had a median overall survival of 17 months, while patients with high hepatic TLG had a median of 10 months. Contrary to Del Carpio et al., they did not find a correlation between SUVmax and patient survival.

## 4. Concluding Remarks

In this systematic review and meta-analysis, we investigated the existing literature to provide comprehensive evidence for the diagnostic and prognostic values of ^18^F-FDG PET in uveal melanoma. Overall, 27 studies were reviewed, and their data were delivered in different sections to help determine the role of ^18^F-FDG PET in different clinical scenarios.

In intra-ocular evaluation, ^18^F-FDG PET was not an accurate imaging modality for detecting primary melanoma in orbit, having a pooled sensitivity of less than 50%. Even considering low SUV thresholds, it seems to be inaccurate for primary tumour detection. However, ^18^F-FDG PET was a highly specific modality for detecting intra-ocular melanoma in cases of suspicion. In terms of the association of ^18^F-FDG uptake and tumour characteristics, the literature mainly supports that the greater the size (e.g., thickness/height, diameter), the higher the avidity. Also, it was shown that the higher percentage decrease in metabolic activity may predict a more favourable response to treatment considering later tumour regression.

Regarding metastasis, overall, ^18^F-FDG PET was a highly reliable modality. Its strength was mainly due to its potential to detect hepatic metastases accurately. Based on our analysis, ^18^F-FDG PET could accurately exclude or confirm hepatic metastasis in uveal melanoma patients (sensitivity and specificity at the patient level were 95% and 100%, respectively). However, it should be noted that this accuracy was not high at the lesion level, and ^18^F-FDG PET may miss subcentimetric hepatic metastases. Thus, clinicians can rely on ^18^F-FDG PET results to evaluate the presence of hepatic metastasis, while to find small metastases, an additional MRI should be performed. Higher availability of PET/MRI may obviate this multi-session imaging in the near future.

Noteworthy, it was shown that ^18^F-FDG PET-missed metastases in the liver might not have a detrimental impact on survival. Patients with undetectable metastases in the liver had significantly higher survivals than those with ^18^F-FDG-positive metastases. Another prognostic value of ^18^F-FDG PET in the liver was that the level of ^18^F-FDG uptake in metastases was a strong predictor of patient survival. Lastly, ^18^F-FDG PET had the potential to characterise lesions from the histopathology perspective. It could distinguish high-risk from low-risk diseases and may provide additional information about the genetic profile of the tumour (e.g., status of chromosome 3).

This study suffered from some limitations. First, the number of studies in the literature was insufficient for a more robust analysis, considering the prognostic value in particular. Second, there were not many studies with negative cases to provide a better understanding of ^18^F-FDG PET specificity within the orbit. Third, as discussed earlier, there were different criteria, particularly in terms of SUV thresholds, to diagnose tumoral lesions. Fourth, data regarding metastatic regions other than the liver were insufficient to perform meta-analytic calculations. Fifth, regarding hepatic metastases, due to insufficiency, we could not provide a lesion-level analysis or size-based calculations to report the corresponding pooled results.

To conclude, although ^18^F-FDG PET is not reliable in detecting primary intra-ocular tumours, it is highly accurate for diagnosing metastatic disease. Also, it is a highly valuable modality in terms of patient prognostication. Hence, ^18^F-FDG PET can be recommended in patients diagnosed with uveal melanoma to enhance decision-making and patient management.

## Figures and Tables

**Figure 1 cancers-16-01712-f001:**
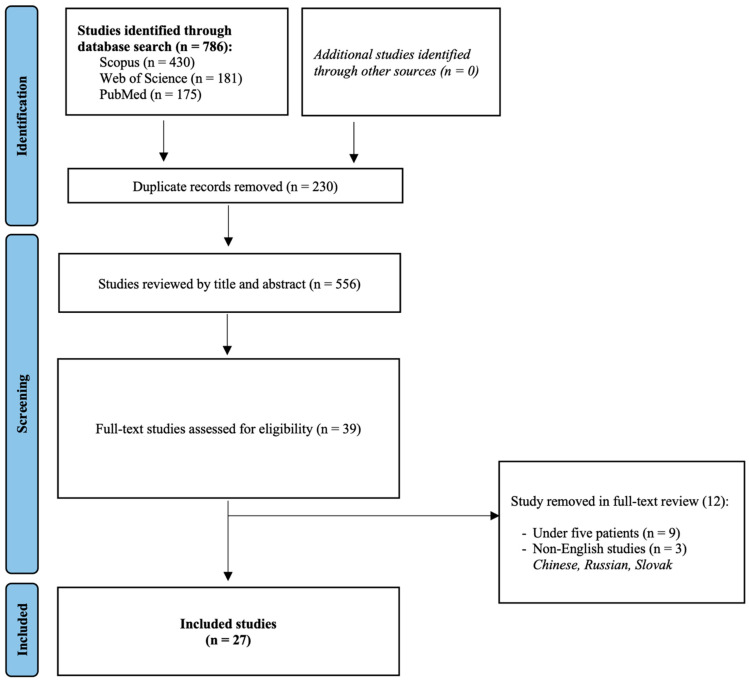
Study selection flowchart.

**Figure 2 cancers-16-01712-f002:**
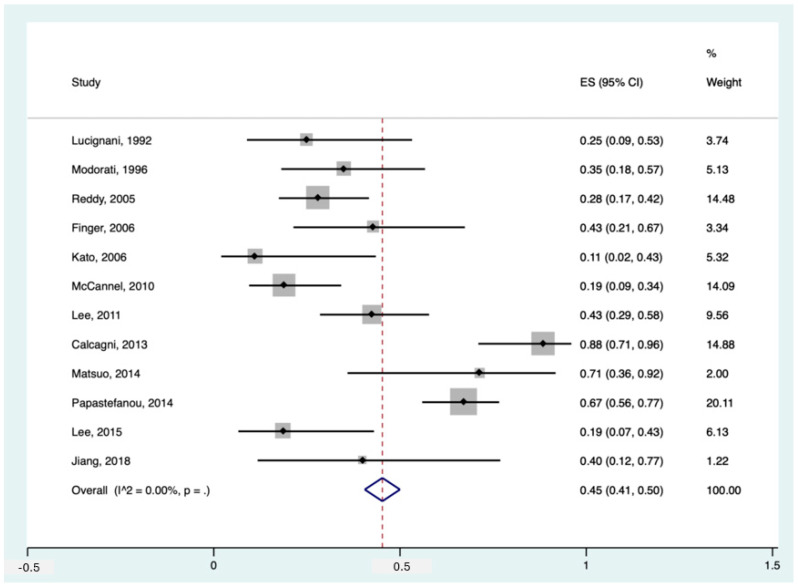
Forest plot for the pooled sensitivity calculation for primary tumour detection. Horizontal lines represent 95% confidence intervals of the individual studies [14,20,24,25,29,31,32,33,34,35,37,38].

**Figure 3 cancers-16-01712-f003:**
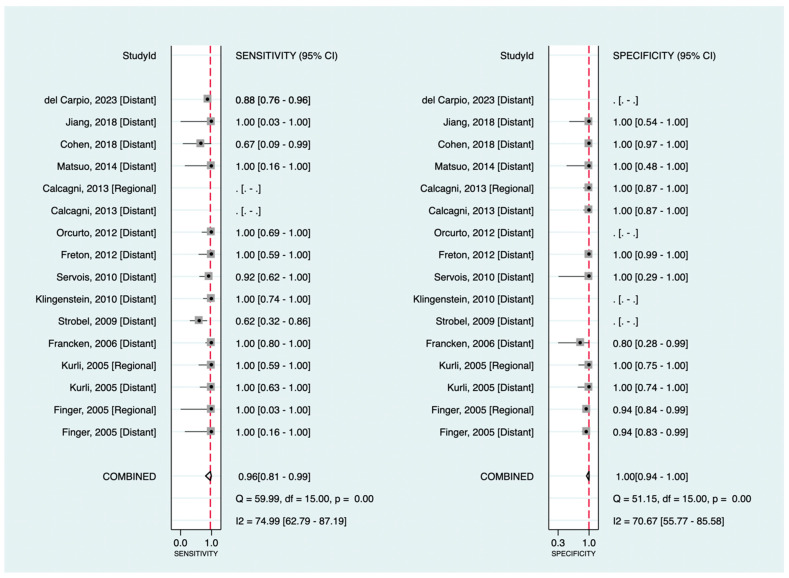
Forest plots for the pooled sensitivity and specificity calculation for detecting metastatic disease. Horizontal lines represent 95% confidence intervals of the individual studies. The dashed red line is the line of summary points across the studies [14,15,16,19,22,23,24,26,28,33,36,39,40].

**Figure 4 cancers-16-01712-f004:**
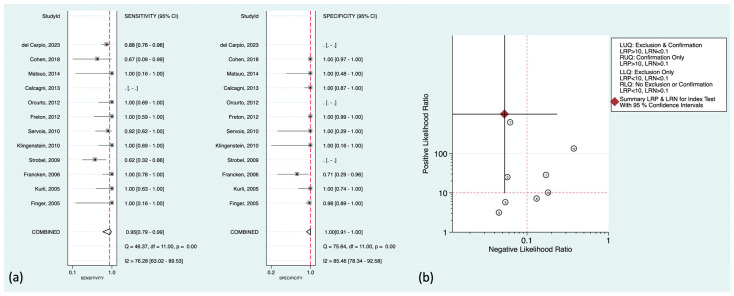
(**a**) Forest plots for the pooled sensitivity and specificity calculation for detecting hepatic metastasis. Horizontal lines represent 95% confidence intervals of the individual studies. The dashed red line is the line of summary points across the studies. (**b**) Likelihood ratio scattergram for liver metastasis assessment. AUC = area under the receiver operating characteristic curve, LLQ = left lower quadrant, LRN = negative likelihood ratio, LRP = positive likelihood ratio, LUQ = left upper quadrant, RLQ = right lower quadrant, RUQ = right upper quadrant, SENS = sensitivity, SPEC = specificity [14,15,16,19,22,23,26,28,33,36,39,40].

**Table 1 cancers-16-01712-t001:** Details of the included studies in the systematic review.

First Author, Year	Modality	Histopathology	Patient Population	Treatment-Naïve Population%	Male%	Mean Age (Years)	Primary Tumour Mean Thickness (mm)	Primary Tumour Mean Diameter (mm)	Primary Tumour Average SUVmax	Metastasis Average SUVmax
Lucignani, 1992 [32]	PET	Uveal Melanoma	12	100%	NA	NA	7	NA	NA	NA
Modorati, 1996 [35]	PET	Uveal Melanoma	20	100%	75%	62	NA	8	NA	NA
Finger, 2005 [19]	PET/CT	Choroidal Melanoma	52	100%	46%	64	NA	NA	NA	NA
Kurli, 2005 [28]	PET/CT	Uveal Melanoma	20	10%	55%	69	NA	NA	NA	NA
Reddy, 2005 [38]	PET/CT	Choroidal Melanoma	50	100%	44%	64	5	11	NA	NA
Finger, 2006 [20]	PET/CT	Choroidal Melanoma - *Epithelioid cell: 4* - *Mixed cell: 6* - *Spindle cell: 4*	14	100%	NA	61	11	19	6.2	NA
Francken, 2006 [22]	PET	Uveal Melanoma	22	NA	41%	55 (median)	NA	NA	NA	NA
Kato, 2006 [25]	PET	Uveal Melanoma	13	100%	62%	60	6	8	NA	NA
Faia, 2008 [18]	PET/CT	Choroidal Melanoma - *Epithelioid cell: 1* - *Mixed cell: 6* - *Spindle cell: 7*	14	100%	57%	62	9	15	5.0	NA
Strobel, 2009 [40]	PET/CT	Uveal Melanoma	13	100%	NA	57	NA	NA	NA	6.0
Klingenstein, 2010 [26]	PET/CT	Uveal Melanoma	12	8%	25%	56 (median)	6	NA	NA	NA
McCannel, 2010 [34]	PET/CT	Choroidal Melanoma	37	100%	59%	64	5	11	NA	NA
Servois, 2010 [39]	PET/CT	Uveal Melanoma	15	100%	47%	56	NA	NA	NA	NA
Finger, 2011 [21]	PET/CT	Choroidal Melanoma	18	100%	22%	65	9	15	3.7	NA
Lee, 2011 [29]	PET/CT	Choroidal Melanoma	40	100%	55%	51	7	11	2.0 (Active + Inactive)	9.6
Freton, 2012 [23]	PET/CT	Choroidal Melanoma	333	100%	49%	64	NA	NA	NA	NA
Orcurto, 2012 [36]	PET/CT	Uveal Melanoma	10	60%	40%	56	8	17	NA	NA
Calcagni, 2013 [14]	PET/CT	Uveal Melanoma - *Epithelioid cell: 7* - *Mixed cell: 10* - *Spindle cell: 9*	26	100%	58%	63	9	13	4.3	NA
Klingenstein, 2013 [27]	PET/CT	Uveal Melanoma	13	0%	31%	57 (median)	NA	NA	NA	NA
Lee, 2014 [30]	PET/CT	Choroidal Melanoma	26	100%	58%	54 (median)	5 (median)	11 (median)	2.6	NA
Matsuo, 2014 [33]	PET/CT	Choroidal Melanoma - *Epithelioid cell: 6* - *Spindle cell: 1*	7	100%	57%	63	8	13	5.4	NA
Papastefanou, 2014 [37]	PET/CT	Uveal Melanoma - *Epithelioid cell: 13* - *Mixed cell: 27* - *Spindle cell: 35*	76	100%	61%	NA	10	15	NA	NA
Lee, 2015 [31]	PET/CT	Choroidal Melanoma	16	100%	56%	54	6	11	NA	NA
Eldredge-Hindy, 2016 [17]	PET/CT	Uveal Melanoma	50	100%	42%	63 (median)	NA	NA	NA	6.9 (median)
Cohen, 2018 [15]	PET/CT	Uveal Melanoma	108	100%	46%	NA	7	13	NA	NA
Jiang, 2018 [24]	PET/CT	Choroidal Melanoma	7	71%	71%	49	8	13	3.3	13.0
Del Carpio, 2023 [16]	PET/CT	Uveal Melanoma	51	100%	59%	62 (median)	NA	NA	NA	8.5 (median)

## Data Availability

All datasets and analyses are available and can be accessed upon a reasonable request from the corresponding author.

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
