# Peer review of "A Systematic Review and Meta-Analysis on the Diagnostic and Prognostic Values of 18F-FDG PET in Uveal Melanoma and Its Hepatic Metastasis"

_cancers, 2024, doi:10.3390/cancers16091712_

Round 1
Reviewer 1 Report
Comments and Suggestions for Authors
Title: A Systematic Review and Meta-Analysis on the Diagnostic and Prognostic Values of 18F-FDG PET in Ophthalmic Melanoma
General Comments:
This manuscript contains a multicenter, international cooperative review on a topical and controversial subject. It deserves a meta-analysis, as multiple case series exist without any formal comparative studies. The review is not comprehensive as it does not address the value of whole-body PET/CT compared to segmental radiographic imaging to detect metastatic melanoma of the skin, bone, brain, and lung. In contrast, it does achieve a comprehensive review and meta-analysis of the positive predictive value of 18F-FDG PET for uveal melanoma metastatic to the liver. Overall, the authors have written and compiled a well-written paper.
In consideration that the authors to not specifically address extrahepatic disease, they should consider changing the title, “ A Systematic Review and Meta-Analysis on the Diagnostic and Prognostic Values of 18F-FDG PET/CT in Ophthalmic Melanoma” to “ A Systematic Review and Meta-Analysis on the Diagnostic and Prognostic Values of 18F-FDG PET/CT for Uveal Melanoma and its Hepatic Metastasis.”
This is because, it is well-known that PET/CT can also detect extra-hepatic metastasis and/or multiple-organ metastases, as the authors are unable to comment on that due to the paucity of published data as mentioned in their lacunae. Therefore, either include and thus review extrahepatic metastasis or clearly define that this work is limited to hepatic metastasis throughout the manuscript. Also, for uniformity, the authors should use either hepatic metastasis or liver metastasis throughout the manuscript. Similarly, they should use the term uveal melanoma instead of ophthalmic melanoma throughout the manuscript, as the latter could include conjunctival and eyelid variants.
Specific comments:
Line 57: That can be, should be “that is widely used.” Though some centers us PET/CT for initial staging, many centers use it for staging when there is evidence of metastasis.
Table 1 is organized by author, in alphabetical order. It would make more functional sense to order them by date published or numbers of cases in each study. Alternatively they could be grouped by function as seen in Figure 1.
3.4, 219-231: The findings that subcentimeter hepatic metastasis might not be avid and thus affect diagnosis is interesting.
3.5, there could be a section describing PET/CT for detection of hepatic versus extrahepatic metastasis as recently published in Garg et. al.
Comments on the Quality of English Languagewell writen in English
Author Response
Distinguished Reviewer #1:
General Comments:
This manuscript contains a multicenter, international cooperative review on a topical and controversial subject. It deserves a meta-analysis, as multiple case series exist without any formal comparative studies. The review is not comprehensive as it does not address the value of whole-body PET/CT compared to segmental radiographic imaging to detect metastatic melanoma of the skin, bone, brain, and lung. In contrast, it does achieve a comprehensive review and meta-analysis of the positive predictive value of 18F-FDG PET for uveal melanoma metastatic to the liver. Overall, the authors have written and compiled a well-written paper.
In consideration that the authors to not specifically address extrahepatic disease, they should consider changing the title, “ A Systematic Review and Meta-Analysis on the Diagnostic and Prognostic Values of 18F-FDG PET/CT in Ophthalmic Melanoma” to “ A Systematic Review and Meta-Analysis on the Diagnostic and Prognostic Values of 18F-FDG PET/CT for Uveal Melanoma and its Hepatic Metastasis.”
This is because, it is well-known that PET/CT can also detect extra-hepatic metastasis and/or multiple-organ metastases, as the authors are unable to comment on that due to the paucity of published data as mentioned in their lacunae. Therefore, either include and thus review extrahepatic metastasis or clearly define that this work is limited to hepatic metastasis throughout the manuscript. Also, for uniformity, the authors should use either hepatic metastasis or liver metastasis throughout the manuscript. Similarly, they should use the term uveal melanoma instead of ophthalmic melanoma throughout the manuscript, as the latter could include conjunctival and eyelid variants.
A: Dear reviewer. We appreciate your comment. Our first intention was to cover various aspects of uveal melanoma. However, as you mentioned, the most prominent literature in this regard is on liver metastases. So, we agree to change the title to be more precise, as you suggested. Also, we agree with the consistent terminology you mentioned and changed the manuscript accordingly.
Specific comments:
Line 57: That can be, should be “that is widely used.” Though some centers us PET/CT for initial staging, many centers use it for staging when there is evidence of metastasis.
A: Agree. Changed.
Table 1 is organized by author, in alphabetical order. It would make more functional sense to order them by date published or numbers of cases in each study. Alternatively they could be grouped by function as seen in Figure 1.
A: We ordered them by authors since we thought it would be easier for readers to find a study in the table. However, we agree that publication date would be a more valid indicator. Thus, based on your comment, we changed its order.
3.4, 219-231: The findings that subcentimeter hepatic metastasis might not be avid and thus affect diagnosis is interesting.
A: Cannot agree more!
3.5, there could be a section describing PET/CT for detection of hepatic versus extrahepatic metastasis as recently published in Garg et. al.
A: We added the mentioned reference. Thanks for your suggestion.

Reviewer 2 Report
Comments and Suggestions for Authors
The manuscript of Mirshahvalad et al « a systematic review and meta-analysis on the diagnostic and prognostic values of 18F-FDG PET in ophthalmic melanoma » suggest that 18F-FDG PET is highly accurate for diagnosing metastatic metastasis of uveal melanoma, and therefore could be a valuable modality for prognosis.
In this study Uveal melanoma (presumably including tumors of the ciliary body and other non choroidal tissues) are mixed with choroidal melanoma that are known to perform liver metastasis. If only choroidal melanoma are taken into account in this study, this modify the efficacy of liver metastasis detection by the 18F-FDG PET?
Author Response
Distinguished Reviewer #2:
The manuscript of Mirshahvalad et al « a systematic review and meta-analysis on the diagnostic and prognostic values of 18F-FDG PET in ophthalmic melanoma » suggest that 18F-FDG PET is highly accurate for diagnosing metastatic metastasis of uveal melanoma, and therefore could be a valuable modality for prognosis.
In this study Uveal melanoma (presumably including tumors of the ciliary body and other non choroidal tissues) are mixed with choroidal melanoma that are known to perform liver metastasis. If only choroidal melanoma are taken into account in this study, this modify the efficacy of liver metastasis detection by the 18F-FDG PET?
A: Dear reviewer. Thanks for the raised point. We agree that this exclusive analysis would help the results. However, the reason we did not opt for that is as you can see, only three studies were exclusively on choroidal melanoma in terms of liver metastasis detection. This issue makes this analysis not really valid for pooling. However, since this was an interesting point to add, we inserted the corresponding findings in the results with a descriptive explanation. We hope this can satisfy your expectations.
